# Point-of-Care Diagnostics for Diagnosis of Active Syphilis Infection: Needs, Challenges and the Way Forward

**DOI:** 10.3390/ijerph19138172

**Published:** 2022-07-04

**Authors:** Minh D. Pham, Jason J. Ong, David A. Anderson, Heidi E. Drummer, Mark Stoové

**Affiliations:** 1Burnet Institute, Melbourne 3004, Australia; david.anderson@burnet.edu.au (D.A.A.); heidi.drummer@burnet.edu.au (H.E.D.); mark.stoove@burnet.edu.au (M.S.); 2Department of Epidemiology and Preventive Medicine, School of Public Health and Preventive Medicine, Monash University, Melbourne 3004, Australia; 3Melbourne Sexual Health Centre, Melbourne 3053, Australia; jong@mshc.org.au; 4Department of Microbiology, Monash University, Melbourne 3800, Australia; 5Peter Doherty Institute for Infection and Immunity, The University of Melbourne, Melbourne 3000, Australia

**Keywords:** syphilis, diagnostics, point-of-care, pregnant women, men who have sex with men

## Abstract

Syphilis, a curable sexually transmitted infection, has re-emerged as a global public health threat with an estimated 5.6 million new cases every year. Pregnant women and men who have sex with men are key target populations for syphilis control and prevention programs. Frequent syphilis testing for timely and accurate diagnosis of active infections for appropriate clinical management is a key strategy to effectively prevent disease transmission. However, there are persistent challenges in the diagnostic landscape and service delivery/testing models that hinder global syphilis control efforts. In this commentary, we summarise the current trends and challenges in diagnosis of active syphilis infection and identify the data gaps and key areas for research and development of novel point-of-care diagnostics which could help to overcome the present technological, individual and structural barriers in access to syphilis testing. We present expert opinion on future research which will be required to accelerate the validation and implementation of new point-of-care diagnostics in real-world settings.

## 1. Background

Syphilis is a bacterial infection caused by spirochaete *Treponema pallidum* subspecies *pallidum* (*T. pallidum*), which can be transmitted by direct sexual contact or vertically from mother to child during pregnancy. The natural course of syphilis includes primary, secondary, latent and tertiary stages, with primary and secondary stages being the most infectious. If left untreated, syphilis can lead to serious complications and permanent damage in the nervous and cardiovascular systems that can be life-threatening [1]. After the advent of penicillin, an effective treatment for syphilis, the prevalence of syphilis was reduced significantly. However, the disease remains an important global health challenge, with an estimated 5.6 million new cases every year [2]. Whilst low-income countries still bear a large burden of the global syphilis cases, an increasing number of new syphilis infections have been reported from developed and emerging economies in several regions, including North America, Europe and the Asia-Pacific [3].

All pregnant women should be tested for syphilis because undiagnosed and untreated syphilis infection during pregnancy could lead to mother to child transmission of syphilis (congenital syphilis), which results in serious health consequences to mothers and their babies [2]. These health problems could be completely resolved by timely diagnosis and adequate treatment with penicillin G benzathine, and therefore the World Health Organization (WHO) recommends syphilis testing for all pregnant women in the first trimester (or at the first antenatal care visit). Treatment for and follow-up of exposed infants born to infected mothers are also recommended to reduce morbidity and mortality associated with congenital syphilis [4].

The current re-emergence of syphilis in high and middle-income countries is mainly driven by epidemics among men who have sex with men (MSM). Existing evidence suggests that syphilis disproportionately affects MSM who live with HIV and those using HIV pre-exposure prophylaxis (PrEP) [5]. The incidence and prevalence of syphilis infection among MSM has increased significantly over the past decades [6]. This is of particular concern because primary syphilis lesions can increase the risk of acquiring and transmitting HIV, whereas HIV can accelerate the natural history of syphilis [7]. One feature of the current syphilis epidemics among MSM is the increasing prevalence of repeat syphilis infections, with increasing proportions of cases being reported from Belgium [8], Australia [9] and the United States [10]. With mounting evidence on the increased frequency and the mostly asymptomatic nature of these cases among MSM [11,12,13,14], frequent syphilis screening after initial diagnosis and treatment has been identified as a key strategy to control the epidemic in this population. National guidelines in Australia [15] and the United States [16] recommend syphilis testing for sexually active MSM as frequently as every three months.

## 2. Challenges in Laboratory Diagnosis of Syphilis

As clinical manifestations of syphilis are highly variable depending on the stage of the disease, laboratory test results for syphilis must be viewed and interpreted in the context of an individual’s medical history to inform the accurate clinical diagnosis and management of the case [17]. The laboratory diagnosis of syphilis relies on direct detection of the pathogen (*T. pallidum*) or serological diagnoses of *T. pallidum* infection. Direct detection methods such as dark-field microscopy (DFM) and nucleic acid amplification testing (NAAT, e.g., polymerase chain reaction/PCR) are particularly useful in diagnosing early primary syphilis when antibodies are not yet detectable. These methods, however, depend on the evidence and suitability of clinical samples such as moist syphilis ulcer/lesions, which can resolve spontaneously and are often not present at the time of clinical visit. These tests also require highly trained laboratory technologists and/or complex/expensive laboratory instruments [18]. Furthermore, test performance (sensitivity and specificity) varies depending on specimen type, stage of infection, and lab technicians’ experience and skills. Therefore, direct detection methods are rarely performed outside of reference laboratories or specialised sexual health services [19].

Serological testing remains the mainstay for diagnosing syphilis infection because of the ease of blood collection (compared to taking samples from syphilis lesions) and the availability and affordability of serological assays. Serology tests for syphilis can be divided into the treponemal antibody test (TT) and the non-treponemal antibody test (NTT); a combination of positive TT and reactive NTT is required for the diagnosis of active (infectious) syphilis infection [20]. But each of the current TT and NTT have their limitations, hampering syphilis control and prevention efforts in real-world settings.

Treponemal tests are available in automated, laboratory-based, as well as point-of-care (POC) assay formats. Evaluation studies have shown that laboratory TTs are highly sensitive and specific for diagnosing syphilis at all stages of infection other than very early primary syphilis [21]. POC assays can achieve diagnostic accuracy levels comparable to laboratory tests in clinical settings [22]. However, treponemal antibodies remain for life even after effective treatment, and TTs cannot distinguish between active/infectious and past treated syphilis infections. NTTs (rapid plasma reagin/RPR or venereal disease research laboratory/VDRL tests) are generally laboratory-based assays performed on serum. They therefore require a return visit to the clinic for test results and clinical management, leading to the potential for patient loss-to-follow up. Furthermore, NTTs have reduced sensitivity in the diagnosis of primary (62–78%) and late latent syphilis (61–75%) [23] and may not provide accurate information about syphilis treatment efficacy because the NTT titre decreases over time even without treatment [24]. The NTT titre could be influenced by HIV infection and the use of antiretroviral therapy (ART) [25]. The interpretation of NTT results is also challenging in the case of minimum reactive and/or low titre test results (e.g., RPR < 1:8). Biological false-positive results can occur in the context of pregnancy, autoimmune diseases or other infections (e.g., malaria, hepatitis C) [26], and NTT may be reactive in areas where yaws, pinta or non-venereal disease is endemic [27]. A RPR test can return a false negative result in early primary, late latent or tertiary syphilis [28], but a false-negative RPR test can also be found in primary or secondary syphilis with very high antibody titre (prozone phenomenon) [29]. As syphilis reinfection is associated with attenuated immunological responses and reduced antibody levels [30], the sensitivity of NTT decreases and test results may not be reliable for diagnosis of repeat syphilis [31].

## 3. The Need for Rapid, Point-of-Care Test to Improve Frequent Testing among Pregnant Women and MSM

Rapid, point-of-care (POC) testing for syphilis has been an integral part of syphilis screening policies and is widely implemented in many countries for the prevention of mother to child transmission of syphilis (congenital syphilis) [4]. However, the global coverage of syphilis screening in pregnancy is still suboptimal. In 27 (33%) of the 81 priority countries that account for >90% of children under-five deaths, less than 50% of pregnant women were tested for syphilis. Only four countries met the WHO target of 95% syphilis testing for all pregnant women and 95% treatment for infected mothers to eliminate congenital syphilis [32]. Although on-site maternal syphilis testing and treatment are cost-effective, low prioritization of syphilis control, limited funding for purchase and inefficient distribution of syphilis test kits contribute to low coverage of syphilis testing in antenatal care in many low and middle-income countries [33]. Improving the availability of a low-cost, accurate and easy-to-use rapid test for syphilis could help to alleviate these barriers and increase the coverage of maternal syphilis testing globally.

The global re-emergence of syphilis among MSM requires novel diagnostic and prevention approaches that encourage timely, accurate diagnosis and treatment of active infections to break the chain of transmission. MSM are at high risk of infection and continue to bear a significant burden of the syphilis pandemic [34]. There are multiple individual and structural barriers to access needed health care services, including testing for syphilis and other related sexually transmitted infections [35]. In many developed countries, the advent of effective ART making HIV untransmissible, and the widespread availability of PrEP, has led to increasing levels of comfort with condomless sex [36] and an associate decrease in condom use among MSM [37,38,39]. In the context of these biomedical HIV prevention approaches, modelling studies of syphilis transmission have emphasised the importance of syphilis control strategies that focus on increased testing among MSM [40]. Rapid POC tests have the potential to play a critical role in the global efforts to control syphilis among MSM by providing more convenient and acceptable care models to improve the coverage and frequency of syphilis testing. Studies among sexually active MSM in Australia [41] and elsewhere [42] indicated a preference for rapid testing over traditional laboratory serology. Decentralized POC HIV and the syphilis testing model was found to be acceptable, providing opportunities to reach MSM who have not been tested or reported infrequent testing [43].

## 4. Challenges in Point-of-Care Diagnostic Landscape for Syphilis and the Way Forward

Most commercially available POC platforms for syphilis, including dual HIV/syphilis assays, are designed to detect treponemal antibodies. Systematic reviews showed good sensitivity and specificity in pregnant women [44] and other populations (e.g., men and nonpregnant women [22]) for both single rapid syphilis [45,46,47] or dual HIV/syphilis POC tests [48]. However, in settings with a high prevalence of previously treated syphilis, treponemal tests are less useful and confirmatory testing is required for a diagnosis of active infection and treatment initiation. One commercially available POC assay provides both treponemal and non-treponemal results and, therefore, in theory, can bypass the need for a syphilis confirmatory test (DPP Syphilis Screen & Confirm, Chembio Diagnostics, Medford, NY, USA) [49]. A meta-analysis of the performance of the DPP showed a good sensitivity (85–99%) and specificity (88–100%) for syphilis, supporting the use of this test in field settings [50]. Field evaluation studies, however, have indicated that the use of this test may result in significant underdiagnosis and undertreatment (up to 48%) of active syphilis infection among MSM in Italy [51] and pregnant women in Burkina Faso [52], increasing risks of onward transmission, and raising questions on the impact and cost-effectiveness of this on-site dual treponemal/non-treponemal testing approach compared to the treponemal testing strategy in these specific contexts. It was evident that the performance of a POC test may vary significantly in different cohorts and/or target populations, taking into consideration the field conditions (e.g., temperature, humidity), human factors (e.g., test operator: clinical staff/laboratory technician) and the quality assurance system in place. However, it is critically important that an optimal balance of sensitivity and specificity is achieved for the introduction of a new POC test in the field. As there is a clear need for reflexive (confirmatory) testing for syphilis, future research on the dual treponemal/non-treponemal testing is recommended to establish whether, where and in which conditions this testing approach would provide the most value for syphilis control and prevention efforts.

Although the potential of a new POC assay (Syphilis IgA Confirmatory test, Burnet Institute, Melbourne, Australia) to distinguish between active and past/treated syphilis has been demonstrated in a low-risk population (pregnant women) [53], further research is needed to examine test performance in diverse clinical settings, and in populations with a high background prevalence of past episodes of syphilis infection, such as MSM. Other POC tests which can differentiate active from past/treated syphilis infections include an immune-filtration device [54] for simultaneous detection of treponemal and non-treponemal antibodies (Span Diagnostics, Surat, India) and a Smartphone dongle triplex test [55] which can detect HIV, treponemal and non-treponemal antibodies (Columbia University, New York, NY, USA). However, these POC assays are currently not commercially available (Table 1).

There is also a need for further research into new biomarkers of acute or probable chronic active syphilis infection to aid in the development and validation of accurate syphilis POC diagnostics. A human-centered approach, in which test developers target the acceptability and usability of tests with end-users/beneficiaries, should be adopted by healthcare providers and researchers when developing and strengthening test designs to ensure that tests are appropriate for the intended use in target settings/populations. Successful adoption and scale-up of novel diagnostics will require evidence from laboratory, clinical evaluations and field implementation trials. To provide high-quality evidence to inform public health policy and clinical practice, evaluation studies of new POC diagnostics for syphilis should use clinically well-characterised samples stratified by the history of syphilis, stages of infections (primary/secondary/latent/tertiary or neurosyphilis), HIV co-infection status (positive/negative) and have a clearly defined “gold standard” or reference for assessing diagnostic performance of the new test. Because of the complexity, biological and immunological challenges in the diagnosis of active syphilis, the reference should include at least two of the following: (1) Clinical diagnosis (clinical data from physical examinations, sexual history/risk behaviour/exposure, history of diagnosis and treatment of syphilis); (2) Direct detection method (e.g., DFM, molecular/NAAT); and (3) Syphilis serology (TT and NTT).

Cost and cost-effectiveness must also be considered in the introduction of a new POC test and/or testing strategy for syphilis, particularly in resource limited settings. Existing evidence suggests that current POC treponemal tests for syphilis [59] and dual HIV and syphilis POC tests [60] are highly cost-effective. Therefore, the adoption and implementation of new POC diagnostic tools for syphilis should only take place after the demonstration of its added value compared to conventional/existing diagnostic approaches through context-specific clinical evaluations and implementation trials. Acceptability and feasibility of the new POC test/testing strategy from both service providers and client/patients’ perspectives should be assessed, and modelling studies are needed to provide data on cost-effectiveness, clinical and public health effects at the population level. These data are crucial for the design, pilot and implementation of a test and treat strategy towards the elimination of congenital syphilis and to control the resurgence of syphilis among MSM.

Syphilis self-testing is another area in the POC diagnostic landscape that needs further research and development. Self-testing enables home-testing models which can address barriers associated with clinic/facility-based testing by offering people the opportunity to have a test at a convenient time and place. There are rapid syphilis screening tests marketed as “home tests” that can be used by the untrained and unsupervised public. However, these multi-component, blood-based tests have identical testing procedures compared to those for professional use, causing concerns about the correct use and interpretation of test results. There is a lack of data on field validation, optimisation of test design and testing procedures to ensure that the diagnostic performance of these tests is maintained in the hands of lay users. The assessment of similarly designed POC tests [61] or devices [62] for HIV revealed that obtaining and transferring the blood (finger-prick) specimen, and handling the buffer were major difficulties from the users’ perspective, whereas an integrated self-test which incorporates all procedural steps into one easy-to-use device can make it easier for the lay end-user to follow instructions for use, reduce the scope for user errors and improve the overall usability of the test [63]. Such an integrated self-test for syphilis would be a welcome development in the current pool of diagnostics for syphilis, offering a “self-screening” opportunity to people at risk of infection, which may help to improve the rates of early diagnosis and the timely treatment of active syphilis.

## 5. Conclusions

The elimination of congenital syphilis and the effective control and prevention of syphilis among MSM require high testing coverage and treatment approaches that are supported by diverse service delivery models with multiple access points for testing. Rapid, equipment-free POC assays for syphilis can offer a range of service delivery models, including putting tests in the hands of the end-user, and can address individual and structural barriers to increase the coverage and frequency of syphilis testing. With the increasing occurrence of repeat and/or past-treated syphilis, no matter which model is used, confirmatory testing should always be part of the testing package. The test should be provided on-site with the result returned and a clinical decision made on the same visit. Toward that end, reliable rapid POC immunoassays for confirmation of active syphilis infection are needed. Simple, easy-to-use self-tests and instrument-free/disposable true POC molecular tests for early detection of syphilis infection are also desirable. Well-designed pre-market evaluations and implementation trials are crucial to provide data on clinical utility, cost-effectiveness and potential public health impacts of such novel POC tests, informing decisions on the adoption and implementation strategies. From the end-user perspective, health preference research should be conducted to understand how target populations (e.g., pregnant women, MSM) want to be tested and how they value each POC testing model. Demonstration projects and implementation research are required to provide information and guidance for integrating cost-effective service delivery models into national healthcare systems.

## Figures and Tables

**Table 1 ijerph-19-08172-t001:** Point-of-care tests for syphilis with sensitivities and specificities from published evaluation studies [44,51,52,53,54,55,56,57,58].

Test	Specimen for Testing/ Sample Type	Test Type/ Target Antibody	Sensitivity	Specificity	Regulatory Approval
Alere Determine Syphilis TP (Abbott Diagnostics, Maidenhead, UK)	Whole blood, serum, plasma	Treponemal	59.6–100	95.7–100	CE marked ^§^
SD Bioline Syphilis 3.0 (Standard Diagnostics, Yongin, South Korea)	Whole blood, serum, plasma	Treponemal	51.4–100	95.5–100	CE marked ^§^
SyphiCheck WB (Qualpro Diagnostics, Verna, Goa, India)	Whole blood, serum, plasma	Treponemal	64–97.6	98.4–99.7	CE marked ^§^
VisiTect Syphilis (Omega Diagnostics, Littleport, Cambridgeshire, UK)	Whole blood, serum, plasma	Treponemal	72.7–98.2	98.1–100	CE marked ^§^
Syphilis Health Check (Diagnostic Direct, Youngstown, OH, USA)	Whole blood, serum, plasma	Treponemal	50–100	50–100	CE marked ^§^ FDA cleared ^†^
SD Bioline HIV/ Syphilis Duo (Standard Diagnostics, Yongin, South Korea)	Whole blood, serum, plasma	HIV	89.4–100	96.3–100	CE marked ^§^ WHO pre-qualified ^‡^
		Treponemal	66.2–100	96–100	
Multiplo rapid TP/HIV antibody test (MedMira, Halifax, Nova Scotia, Canada)	Whole blood, serum, plasma	HIV	93.8–97.9	94.2–100	CE marked ^§^
		Treponemal	81–94.1	94.2–100	
INSTI Multiplex HIV-1/HIV-2/Syphilis Antibody Test (bioLytical Lab, Richmond, BC, Canada)	Whole blood, serum, plasma	HIV	98.8–100	95.5–100	CE marked ^§^
		Treponemal	56.8–87.4	97–98.5	
Chembio DPP HIV/ Syphilis (Chembio Diagnostic Systems, Hauppauge, NY, USA)	Whole blood, serum, plasma	HIV	90.6–100	97.2–99.6	CE marked ^§^ FDA cleared ^†^ WHO pre-qualified ^‡^
		Treponemal	47.4–98.8	97–100	
Chembio DPP HIV-HCV-Syphilis Assay (Chembio Diagnostic Systems, Hauppauge, NY, USA)	Whole blood, serum, plasma	HIV	95.7–100	99.7–100	CE marked ^§^
		HCV	91.8	99.3	
		Treponemal	44–52.7	98.7–99.6	
Chembio DPP Syphilis Screen & Confirm (Chembio Diagnostic System, Hauppauge, NY, USA)	Whole blood, serum, plasma	Treponemal	65.4–98.2	91.2–100	CE marked ^§^
		Non-treponemal	46.1–98.2	89.4–100	
DPP (Span Diagnostics Ltd, Surat, India)	Whole blood, serum, plasma	Treponemal	97.3	99.1	Not commercially available
		Non-treponemal	96.5	97.7	
Smartphone dongle triplex test (Columbia University, New York, NY, USA)	Whole blood, serum, plasma	HIV	100	87	Not commercially available
		Treponemal	92	92	
		Non-treponemal	100	79	
TP-IgA test * (Burnet Institute, Melbourne, Australia)	Whole blood, serum, plasma	Treponemal	62.7	98–99.6	Not commercially available
		Non-treponemal	96.1–100	84.7–99.4	

* The TP-IgA test detects Treponemal (TP) Immunoglobulin A (IgA) specific antibody with an intended use as a confirmatory test (after a positive result on a rapid total Treponemal antibody test e.g., Alere Determine Syphilis TP) for diagnosis of active syphilis infection. ^§^ CE marked: European Conformity marking for in-vitro medical diagnostics; ^†^ FDA: US Food and Drug Administration; ^‡^ WHO pre-qualification: World Health Organization marketing authorisation.

## Data Availability

Not applicable.

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
