# Peer review of "Point-of-Care Diagnostics for Diagnosis of Active Syphilis Infection: Needs, Challenges and the Way Forward"

_ijerph, 2022, doi:10.3390/ijerph19138172_

Round 1

Reviewer 1 Report

Line 41: which resulted in serious health" - correct the grammar

Lines 53 - 57: too frequent repetition of "repeat syphilis infections". Please rephrase.

Line 80: Clinically, it is easier to obtain samples from syphilitic lesions than venepuncture. The issues are that the lesions resolve spontaneously and are usually not present at the time of the medical visit; and, as mentioned; the lack of expensive equipment and skilled lab staff to perform and interpret the direct tests.

Line 146: please list the name and manufacturer of this assay along with references of relevant studies.

Line: 150: what is meant by "high misclassification'? That is not a term used for reading of tests nor interpretation of tests.

Line 151: please list the name and manufacturer of this assay

There is no detailed review of the literature on syphilis POCTs. A description of the different ones available and their sensitivities and specificities.

If this is not available, then this manuscript is better suited to note of letter; there is no new information added to the literature and just re-iterates what is mentioned in about a 100 other articles on this topic.

Author Response

Thank you. We have responded to all reviewers' comments in the file attached

Reviewer 2 Report

This is an article discussing the role of point of care testing in addressing the resurgence of syphilis especially among vulnerable groups. The article is relevant and has discussed the nuances of both treponemal and non-treponemal tests.

I have minor comments/suggestions:

Comments:

Line 146: One commercially available POC assay provides both treponemal and non-treponemal results and therefore, in theory, can bypass the need for a syphilis confirmatory test. – Please provide citation and specify the POC assay.

Please include a discussion on the cost of syphilis testing and feasibility in resource-limited settings. One study in the USA showed “rapid syphilis tests were conducted on 595 people at an average cost of $213 per person. Twenty-three cases of syphilis were confirmed and treated for an average cost of $5517 per case, ranging from $3604 at a rehabilitation facility to $13,140 at LGBTQ venues served by a mobile clinic. (PMID: 33993158)” This might be too costly for public health programs in developing countries. If there are available and cheap POC tests, please include them in the article.

Please comment on the rapid (10-minute) fingerstick treponemal-based antibody test called the Syphilis Health Check (SHC) -- where the sensitivity of the test has been lower in post-marketing studies compared with the FDA trial studies. In a post-FDA approval evaluation of the SHC (PMID: 27787496; Matthias et al, 2016), the sensitivity and specificity of the test were found to be 71.4 and 91.5 percent (lower than the >95 percent sensitivity and specificity that reported by the manufacturer).

If feasible, suggest to include a table of approved rapid syphilis tests with sensitivity and specificity (with notes whether these are available in developing countries).

Author Response

(The authors gave the same response as above.)

Reviewer 3 Report

1) There are several reviews on this topic. What the new information the authors are providing in this commentary is not clear.

2) Authors need to provide currently available Lab and POC tests and their merits and demerits in tabular form.

3) Tests for screening, diagnosis, and prognosis to be given separately and their merits and demerits.

4) Challenges in each step of POC to be given in a diagram including the logistics etc and how they can be overcome.

5) Is there any genetic predisposition test for syphilis is possible?

Over all, the article is not thought provoking and what the futuristic plans to be considered for POC in syphilis is not much conceived. 

Author Response

(The authors gave the same response as above.)

Round 2

Reviewer 1 Report

God job on addressing the identified deficiencies.